# Development and internal validation of the patient safety experience scale for inpatients

**On-Jeon Baek[1]◉, Sun-Hwa Shin◉[2]◉\***

1 Department of Nursing, Eulji University College of Nursing, Gyeonggi-do, Korea, 2 Department of Nursing, Sahmyook University College of Nursing, Seoul, Korea

◉ These authors contributed equally to this work.
* shinsh@syu.ac.kr

## Abstract

There is an increasing need for a practical instrument that captures patient safety experiences from the inpatient perspective and is suitable for clinical application. This study aimed to develop a Patient Safety Experience Scale (PSES) reflecting inpatient safety indicators and to evaluate its reliability and validity. An initial pool of 90 items was generated through a literature review and qualitative interviews, from which 60 items were selected based on expert evaluation and content validity assessment. A survey was conducted among 549 inpatients. Data were analyzed using item analysis, confirmatory factor analysis, Pearson's correlation, Cronbach's alpha, and intraclass correlation coefficients (ICC) using SPSS 26.0 and AMOS 21.0. The final scale comprised 30 items across six factors: patient identification, prevention of medication errors, fall prevention, infection prevention, compliance with safety in daily life, and information sharing. The PSES demonstrated excellent internal consistency (Cronbach's $\alpha = .95$) and strong test–retest reliability (ICC = .89). Additionally, it showed strong concurrent validity with the patient participation scale, with a correlation coefficient of.91. These findings support the internal validity of the PSES as a reliable and feasible instrument for systematically assessing safety experiences of inpatients. This scale may facilitate targeted quality improvement efforts and contribute to fostering a patient-centered safety culture in healthcare settings.

## 1. Introduction

### 1.1. Background

Patient safety is a key factor in determining the quality of medical services by minimizing unnecessary harm. The World Health Organization defines it as reducing risks associated with healthcare to an acceptable minimum level [1–3]. The Korean Nurses' Code of Ethics states that patient safety should be given top priority in the nursing process [4]. The Korean government is also strengthening patient safety policies and systematically evaluating patient safety incidents and prevention activities

**Data availability statement:** All relevant data are within the paper and its Supporting Information files.

**Funding:** The author(s) received no specific funding for this work.

**Competing interests:** The authors have no conflicts of interest.

through administrative data and reporting systems [5]. However, to date, patient safety evaluations have mainly focused on medical professionals, and there have been limitations in confirming safety incident experiences from the patient's perspective [1]. This highlights the need to evaluate and reflect on the safety experiences of patients and develop patient-centered safety evaluation instruments.

Recently, the Organization for Economic Cooperation and Development (OECD) has made patient-centeredness a primary goal and has developed Patient-Reported Incidence Measures (PRIMs), which enable patients to directly participate in reporting safety indicators during the provision of healthcare services. Through these efforts, the OECD seeks to systematically assess and improve patient safety experience [6]. "Patient safety experience" is a concept that includes safety-related preventive activities, communication with healthcare providers, and responses to incidents as directly perceived and recognized by patients [6]. To evaluate this, various indicators have been proposed, including the OECD's PRIMs, Agency for Healthcare Research and Quality (AHRQ), Patient Safety Indicators (PSIs) in the United States, Joint Commission International (JCI), International Patient Safety Goals (IPSG), and safety indicators from the Korea Institute for Healthcare Accreditation [6–9]. By measuring the safety experiences of inpatients based on these indicators, it is possible to encourage patient participation in patient safety, prevent safety incidents, and use the results as important data for evaluating and improving patient safety activities [10].

In response to the OECD's recommendation that there was a lack of clear mechanisms to ensure patient safety, the Health Insurance Review and Assessment Service in Korea has developed a standard patient experience assessment questionnaire [11,12]. However, although patient experience assessment is regarded as an important attempt to evaluate the safety of healthcare services from the patient's perspective, it has limitations in specifically identifying the patient safety indicators presented by the OECD, AHRQ, and JCI [1]. To date, patient safety measurement instruments developed for inpatients include patient measures of safety [13], patient-reported experiences and outcomes of safety in primary care [14], patient safety perception [15], patient safety knowledge [16], patient participation [17], and performance of patient safety activities [18]. Among these, the patient safety activity performance instrument developed by Kim and Park [18] includes some attributes of patient safety activities based on the AHRQ patient safety indicators. However, other existing instruments are limited to certain aspects of patient safety. In addition, they do not reflect comprehensive attributes such as experiences of participating in safety incident prevention activities, providing information, and proactive safety activities. Therefore, to specifically assess the level of perception of patient safety among inpatients, it is necessary to develop items based on patient safety indicators and create a scale for safety experiences that are directly reported by inpatients.

Considering the limitations of existing instruments, this study aims to develop a scale that reflects the safety experiences of inpatients and is easy to use. To achieve this, we reviewed previous literature and conducted in-depth, repeated explorations of the components of safety experience among inpatients to derive the concept of patient safety experience. Through this study, by systematically evaluating patient

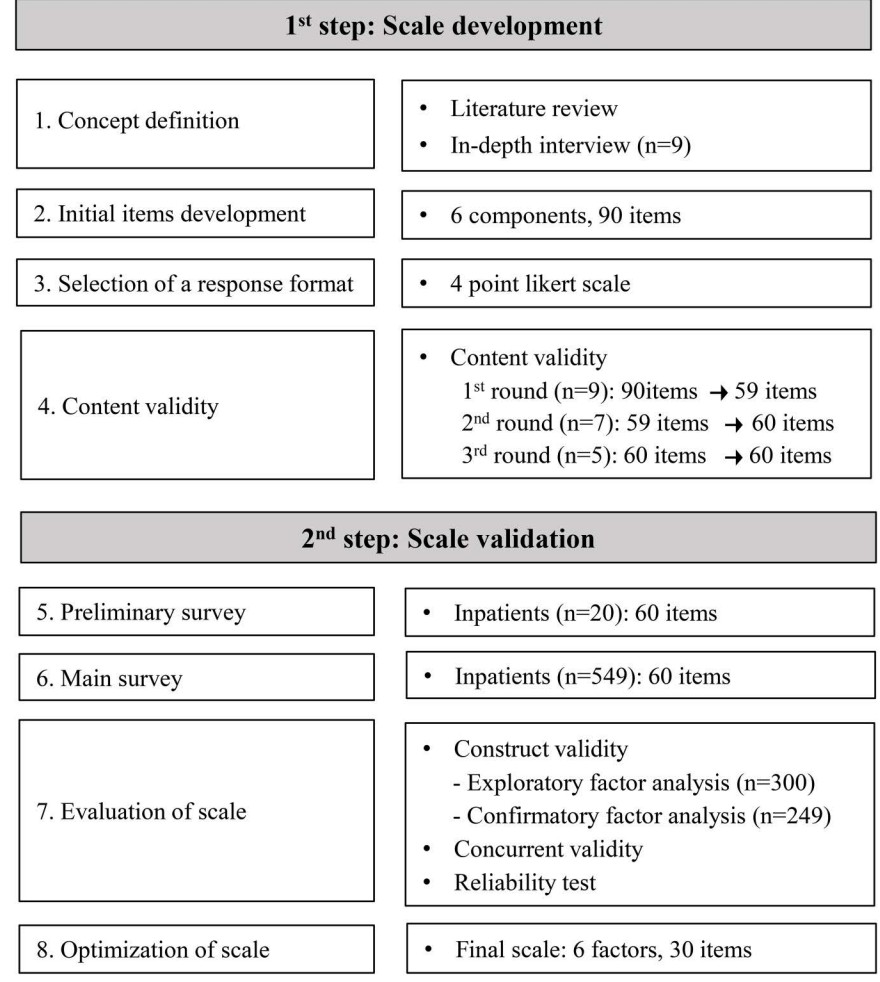 PLOS One

safety experiences, we sought to enhance the understanding of patient safety experiences and ultimately contribute to promoting the participation of inpatients in safety activities.

### 1.2. Aims of the current study

This study aimed to develop a Patient Safety Experience Scale (PSES) that reflected patient safety indicators for inpatients. The specific objectives were to (a) develop a PSES and (b) verify its reliability and validity.

## 2. Methods

### 2.1. Research design

This methodological study aimed to develop a scale to measure patient safety experiences and assess its reliability and validity. This study follows the instrument development and validation procedures proposed by DeVellis and Thorpe [19]. The methodological steps are illustrated in Fig 1.

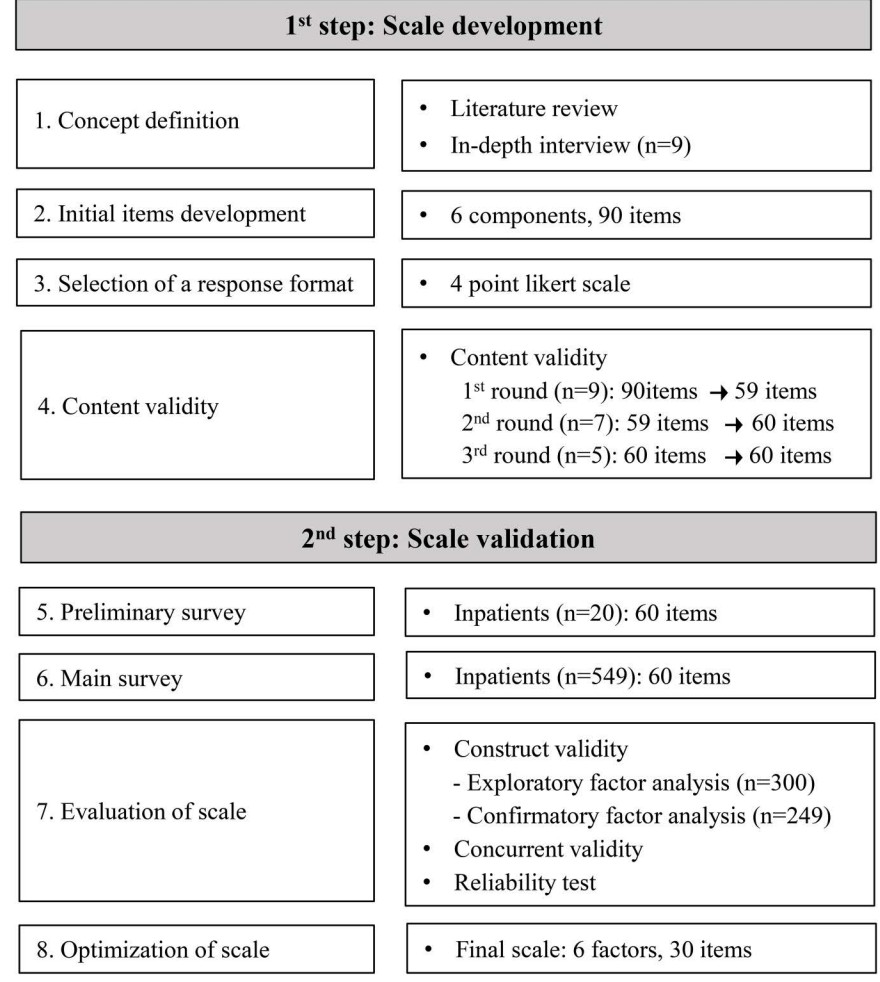

**Fig 1. Flow of the study.**

## 2.2. Scale development step

### 2.2.1. Constructing a concept definition.

To construct a conceptual framework for the patient safety experience, comprehensive literature and in-depth interviews with inpatients were conducted to identify its key components. The literature review focused on publications between January 1, 2012, and October 31, 2023, beginning in 2012 when the OECD initiated the development of patient-reported patient safety indicators [20]. International literature was retrieved using the keywords "patient safety," "experience," "scale," "tool," "instrument," and "hospital" through databases such as PubMed, CINAHL, Web of Science (WoS), Google Scholar, and Embase. Korean literature was searched using the keywords "patient safety," "experience," "tool," "measurement," "hospital," and "medical institution" through RISS, NDSL, KMbase, and DBpia. Based on the inclusion criteria, five international and eight Korean studies that addressed the concept and attributes of inpatient patient safety experiences were selected. Thirteen relevant studies were reviewed to derive the core attributes of the patient safety experience.

In-depth interviews were conducted to gain a comprehensive understanding of inpatients' experiences related to patient safety. Participants were purposively selected to ensure diversity in characteristics such as age, sex, medical department, length of hospital stay, and disease severity, including the inclusion of high-risk groups. The inclusion criteria were as follows: age between 19 and 65 years, history of at least two hospitalizations, understanding the study's purpose, and prior consent to participate. Nine inpatients were interviewed and recruited from five small and medium-sized hospitals (with fewer than 300 beds) located in Seoul and the surrounding metropolitan area.

Nine participants (six women and three men) were interviewed, with an average of 2.6 prior hospitalizations. The interviews began with the question, "What do you think are the safety factors experienced in medical institutions?" Additional questions included, "During your hospitalization, what actions or environments provided by the medical staff or hospital made you feel safe?", "What activities or actions did you perform to ensure your safety during hospitalization?", and "Were there any factors that hindered your safety during hospitalization?" Each interview lasted between 35 and 65 min. The collected data were analyzed using Krippendorff's content analysis method [21].

Based on a comprehensive literature review and in-depth interviews, six key components of patient safety experience were identified: patient identification, prevention of medication errors, fall prevention, infection prevention, compliance with safety in daily life, and information sharing. In this study, patient safety experience is defined as "safety-related experiences that are directly perceived and recognized by inpatients from their own perspective during the process of receiving medical care."

### 2.2.2. Initial items development and selection of a response format.

Initial items were developed based on the conceptual attributes of the patient safety experience through literature review and in-depth interviews. According to Devellis and Thorpe [19], the more initial items, the better, and they should be more than 50% more than the number of questions in the final scale. Referring to this recommendation, this study composed the number of initial items by component from 12 to 18. A total of 90 items were generated, consisting of 12 items for patient identification, 14 for the prevention of medication errors, 16 for fall prevention, 16 for infection prevention, 14 for compliance with safety in daily life, and 18 for information sharing. A 4-point Likert scale suitable for measuring subjective perceptions was used to assess opinions, beliefs, and attitudes [19]. To avoid central tendency bias, the scale excluded neutral options and consisted of the following response categories: strongly disagree (1 point), disagree (2 points), agree (3 points), and strongly agree (4 points) [22]. Higher scores indicated a greater level of patient safety experience, as perceived by inpatient respondents.

### 2.2.3. Content validity.

Content validity was assessed by convening a panel of experts to evaluate the appropriateness of the initial items. Each item was rated on a 4-point Likert scale, and experts provided qualitative feedback regarding the need to modify, supplement, add, or delete specific items. The item-content validity index (I-CVI) was calculated by dividing the number of experts who rated an item as "appropriate" or "very appropriate" by the total number of experts. Items with an I-CVI of.78 or higher were retained. Additionally, the scale-content validity index/average

(S-CVI/Ave) was calculated by averaging the I-CVI scores across all items with a threshold of.90 or higher, indicating acceptable content validity at the scale level [22].

The first round of the content validity assessment was conducted between June 28 and July 9, 2024. The expert panel consisted of nine professionals: two quality improvement (QI) team managers from general hospitals, two nurse managers, three nursing professors with expertise in instrument development, one professor specializing in nursing administration, and one professor of psychology. The I-CVI coefficients for the individual items ranged from 0.67 to 1.00, and the overall S-CVI/Ave was.90. Based on this analysis, 13 items with an I-CVI below.78 and 18 items with redundant or overlapping content were removed. In total, 31 items were deleted, and 47 were revised according to expert feedback, resulting in 59 items being retained for the next stage of validation.

The second content validity assessment was conducted between July 21 and July 25, 2024. The expert panel consisted of seven members: six who participated in the first assessment and one additional professor specializing in fundamental nursing. The I-CVI for the 59 items ranged from.71 to 1.00, and the S-CVI/Ave was.95. Based on the results, two items, one from the fall prevention domain and one from the information sharing domain, were removed because the I-CVI values were below.78. In addition, three new items were added to the infection prevention domain, based on expert recommendations. Following the second assessment, a Korean linguist reviewed the items for grammar, vocabulary, spelling, and sentence clarity, resulting in revisions to the wording of 11 items.

The third content validity assessment was conducted on August 5–14, 2024. The expert panel comprised five members who participated in the second assessment and provided feedback on item revisions. The I-CVI ranged from.80 to 1.00, and the S-CVI/Ave was.99. Because all items met the validity criteria, no further deletions or modifications were made. Consequently, a final total of 60 items were selected for the scale.

**2.2.4. Pilot survey.** A pilot survey was conducted on August 17–24, 2024. To assess the comprehensibility and difficulty of the 60 selected items, a survey was conducted with 20 adults who had been hospitalized in the past year. Among the respondents, seven (35%) were men and 13 (65%) were women, with an average age of 36.3 (±10.55) years. Sixteen (80%) participants reported no underlying diseases. Regarding the number of hospitalizations, five (25%) were hospitalized once, 11 (55%) were hospitalized two to three times, and four (20%) were hospitalized four to five times.

The survey evaluated the font size, sentence comprehensibility, item difficulty, and time required for completion. The participants were encouraged to comment on any items that they found difficult to understand or required clarification. The average time to complete the questionnaire was 10 min. The mean scores for comprehensibility, difficulty, and item length appropriateness, which were rated on a 5-point Likert scale, were 4.35 (±.67), 4.25 (±.79), and 4.15 (±.75), respectively. As no additional suggestions for revision were received, 60 items were finalized and used in the main survey.

## 2.3. Scale validation stage

**2.3.1. Participant and data collection for the main survey.** Participants in the main survey were conveniently sampled from five small- and medium-sized hospitals (each with fewer than 300 beds) located in Seoul city and the surrounding metropolitan area. The inclusion criteria were as follows: In order to secure internal validity by targeting a group with relatively uniform response ability, patients aged 19–65 years, patients hospitalized for at least 3 days, able to understand the purpose of the study, and who provided informed consent. Exclusion criteria included patients hospitalized on the same day, patients visiting the outpatient clinic, and those with cognitive problems or inability to communicate. In total, 550 inpatients were surveyed. This number was determined based on guidelines recommending 5–10 participants per item for exploratory factor analysis (EFA) [19], and 200–400 participants for confirmatory factor analysis (CFA) using structural equation modeling. 10% was added to account for potential dropouts [23].

Data for the main survey were collected between August 26 and September 18, 2024. The researcher visited each participating hospital to explain the study's purpose and procedures to the head nurse or nursing manager and request institutional cooperation. The survey was administered in both paper-based and online formats, depending on participants'

preferences. The research assistants—nurses who were affiliated with each hospital—were trained in advance regarding the study's objectives and data collection procedures.

Participant recruitment was facilitated through posted notices in the hospitals. Individuals who expressed an interest were included in the survey. In the paper-based version, the research assistant explained the study's background, objectives, participation, withdrawal procedures, and potential risks and benefits. After obtaining written informed consent, questionnaires were distributed and collected. The research assistant sent an online survey link to the participants. Upon accessing the link, participants reviewed the "Research Participant Information" and indicated informed consent by selecting "Understood" and "I agree."

**2.3.2. Construct validity.** Construct validity was examined using SPSS (version 26.0) and AMOS (version 21.0 programs (IBM Corp., Armonk, NY, USA). For item analysis, the mean, standard deviation, skewness, kurtosis, item-total correlation coefficient for each item, and the reliability coefficient (Cronbach's $\alpha$) were computed when removing items. EFA was conducted by confirming the suitability of the data using the Kaiser–Meyer–Olkin (KMO) test and Bartlett's test of sphericity. EFA was performed using the maximum likelihood method, as the assumption of normality was satisfied in the item analyses, making it a statistically robust approach. To allow for correlations among latent factors, Promax rotation—an oblique rotation technique widely used in large datasets—was applied, which generates results based on initial orthogonal rotation outcomes [21]. Items with factor loadings below.50 or communalities below.30 were considered for deletion [24].

CFA was conducted to evaluate the model fit. The model was considered acceptable when the ratio of the chi-square ($\chi^2$) value to degrees of freedom was 3 or less. Goodness-of-fit thresholds were defined as follows: SRMR $\leq 0.08$, RMSEA $\leq 0.08$ (with a 90% confidence interval upper bound $\leq 0.10$), CFI $\geq 0.90$, and TLI $\geq 0.90$. The final model showed acceptable fit indices: SRMR $= 0.06$, RMSEA $= 0.07$ (90% CI: 0.05–0.09), CFI $= 0.92$, and TLI $= 0.91$ [23]. Standardized factor loadings ($\beta$) and modification indices (MI) were reviewed to revise the model, if necessary. To verify the convergent validity, the following criteria were applied: standardized factor loading ($\beta$) $\geq 0.50$, average variance extracted (AVE) $\geq 0.50$, and construct reliability (CR) $\geq 0.70$ [25]. Discriminant validity was assessed using the confidence interval of the correlation coefficient ($\Phi \pm 2.00 \times SE$); if the interval did not include 1.00, the constructs were considered distinct, indicating that discriminant validity was established [26].

**2.3.3. Concurrent validity.** Concurrent validity was assessed by examining its correlation with the patient participation scale developed by Song and Kim [27], which addressed the components of patient safety. The scale is a validated and reliable instrument consisting of 21 items across four key subdomains of patient participation: sharing information and knowledge (eight items), participation in the decision-making process (two items), engagement in proactive self-management activities (seven items), and establishing a mutually trusting relationship (four items). Given that patient participation is a core component of the patient safety experience, the conceptual relationship between the two instruments supports the rationale for testing concurrent validity. Responses to the patient participation scale were rated on a 5-point Likert scale ranging from 1 (strongly disagree) to 5 (strongly agree), with higher scores indicating greater levels of patient participation during hospitalization. Cronbach's $\alpha$ of the scale was.92 in the original study [27] and.94 in the present study, demonstrating excellent internal consistency.

**2.3.4. Reliability test.** The homogeneity of reliability was verified by calculating Cronbach's $\alpha$. Stability was assessed by calculating the intraclass correlation coefficients (ICC) to confirm test–retest reliability. For the test–retest, 31 participants from the main survey who voluntarily agreed to participate after being informed of the purpose and procedures of the survey completed the same questionnaire two weeks later [19].

## 2.4. Ethical considerations

Prior to data collection, approval was obtained from the Institutional Review Board (IRB No: SYU 2023-12-008-003). All participants in the in-depth interviews, pilot survey, and main survey were provided with an explanation document that

included details of the purpose and methods of the study, voluntary consent, decision to participate, benefits of participation, information about the researchers, confidentiality, the possibility of withdrawal at any time, and procedures for data storage and disposal. This information was thoroughly explained to the participants. Participants in the online survey were provided with a participant information sheet that included statements that the computerized survey data would be encrypted to ensure security and accessibility only to the researcher, that anonymity would be guaranteed, and that data would be permanently deleted after the completion of the study. Participants who completed and submitted the survey received a token of appreciation, and those who responded to the test–retest received two tokens of appreciation.

## 3. Results

### 3.1 General characteristics of participants

The survey took approximately 15–20 min to complete. In total, 551 participants responded: 149 via paper-based survey and 402 via online survey. After excluding two incomplete responses, data from 549 participants were included in the final analysis.

Among the 549 inpatients who participated in the main survey, 183 (33.3%) were men and 366 (66.7%) were women, with a mean age of 40.4 (±11.53) years. The largest age group comprised those in their 30s (37.2%). In terms of marital status, 53.2% were married and 46.8% were unmarried. Most participants (77.1%) completed university education, and 76.1% were employed. Most (63.6%) reported no underlying diseases, whereas 79.2% had undergone procedures or surgeries. Regarding hospitalization history, 47.7% had been hospitalized 2–3 times, and the most common department of admission was the internal medicine department (57.6%). The most frequent length of hospital stay was 6–15 (51.4%) days. Additionally, 85.4% had no experience of safety incidents, and 73.6% had received patient safety education. No significant differences in general characteristics were observed between randomly assigned EFA (n = 300) and CFA (n = 249) participants (Table 1).

### 3.2. Construct validity

Item means ranged from 3.14 to 3.50. Skewness (from −1.39 to −0.59) and kurtosis (from −0.44 to 2.11) were within acceptable limits, indicating normality. The item-total correlation coefficients ranged from.49 to.69. Reliability analysis showed a Cronbach's α of.97 for all 60 items, which remained unchanged even when any item was removed; thus, all 60 items were initially retained. As a result of the EFA, the KMO test of sampling adequacy was.95, indicating excellent suitability for factor analysis. Bartlett's test of sphericity was also significant ($\chi^2$ = 11,178, df = 1,770, p < .001), confirming the appropriateness of the correlation matrix. A one-factor solution was also supported. Five items with communalities below.30 were removed. The revised EFA, conducted with the remaining 55 items, showed communalities ranging from.30 to.47; and all factor loadings were above.50 (range:.50–.71), indicating satisfactory item loadings on the extracted factor.

As a result of the CFA, the initial model showed a $\chi^2$/df value of 2.17, which was within the acceptable range (<3). However, the model fit indices did not meet the recommended thresholds (CFI = .77, TLI = .76, SRMR = .057, RMSEA = .069, 90% CI [.065,.072]). To improve the model fit, standardized factor loadings (β) and MI were reviewed [23]. Items with standardized loadings below.40 were removed. In addition, items with high MI values between the measurement variables and error terms were eliminated if they overlapped conceptually with other items or were prone to varied interpretations by respondents. Iterative model refinement was performed using the maximum likelihood estimation method, resulting in the removal of 26 items. The final model comprised 30 items across six factors: Factor 1 (five items), Factor 2 (five items), Factor 3 (seven items), Factor 4 (five items), Factor 5 (four items), and Factor 6 (four items). The revised model demonstrated acceptable fit indices: $\chi^2$/df = 1.70, CFI = .91, TLI = .90, SRMR = .047, and RMSEA = .053 (90% CI [.046,.060]).

**Table 1. General characteristics of the participants (N = 549).**

| Characteristics | Categories | Total (n = 549) N (%) | EFA (n = 300) N (%) | CFA (n = 249) N (%) | t/ χ² (p) |
|---|---|---|---|---|---|
| Sex | Man | 183 (33.3) | 99 (33.0) | 84 (33.7) | 0.18 (.856) |
| | Woman | 366 (66.7) | 201 (67.0) | 165 (66.3) | |
| Age (year) | 19~29 | 90 (16.4) | 55 (18.3) | 35 (14.1) | 0.13 (.898) |
| | 30~39 | 204 (37.2) | 101 (33.7) | 103 (41.4) | |
| | 40~49 | 150 (27.3) | 82 (27.3) | 68 (27.3) | |
| | 50~59 | 47 (8.6) | 32 (10.7) | 15 (6.0) | |
| | ≧60 | 58 (10.6) | 30 (10.0) | 28 (11.2) | |
| Marital status | Married | 292 (53.2) | 149 (49.6) | 143 (57.4) | 1.81 (.070) |
| | Single | 257 (46.8) | 151 (50.3) | 106 (42.6) | |
| Education level | Middle school | 17 (3.1) | 8 (2.7) | 9 (3.6) | −0.04 (.970) |
| | High school | 109 (19.9) | 62 (20.7) | 47 (18.9) | |
| | College | 423 (77.1) | 230 (76.6) | 193 (77.5) | |
| Job | No | 131 (23.9) | 74 (24.7) | 57 (22.9) | −0.49 (.628) |
| | Yes | 418 (76.1) | 226 (75.3) | 192 (77.1) | |
| Underlying disease | No | 349 (63.6) | 186 (62.0) | 52 (20.9) | 0.84 (.402) |
| | Yes | 200 (36.4) | 114 (38.0) | 197 (79.1) | |
| Procedure/Surgery | No | 114 (20.8) | 62 (20.7) | 52 (20.9) | 0.06 (.950) |
| | Yes | 435 (79.2) | 238 (79.3) | 197 (79.1) | |
| Hospitalization (number) | 1 | 190 (34.6) | 99 (33.0) | 91(36.5) | −0.06 (.950) |
| | 2~3 | 262 (47.7) | 152 (50.7) | 110 (44.2) | |
| | 4~5 | 58 (10.6) | 30 (10.0) | 28 (11.2) | |
| | 6~9 | 24 (4.4) | 10 (3.3) | 14 (5.6) | |
| | ≧10 | 15 (2.7) | 9 (3.0) | 6 (2.4) | |
| Medical department | Medicine | 316 (57.6) | 178 (59.3) | 138 (55.4) | −1.12 (.263) |
| | Surgery | 98 (17.9) | 54 (18.0) | 44 (17.7) | |
| | Rehabilitation | 21 (3.9) | 10 (3.4) | 11 (4.4) | |
| | Orthopedics | 76 (13.8) | 38 (12.7) | 38 (15.3) | |
| | Others | 38 (6.9) | 20 (6.7) | 18 (7.2) | |
| Length of hospital stay (day) | ≤5 | 208 (37.9) | 116 (48.7) | 92 (36.9) | −0.87 (.383) |
| | 6~15 | 282 (51.4) | 155 (51.7) | 127 (51.0) | |
| | ≥16 | 59 (10.7) | 29 (9.7) | 30 (13.0) | |
| Patient safety incident | No | 469 (85.4) | 257 (85.7) | 212 (85.1) | −0.17 (.862) |
| | Yes | 80 (14.6) | 43 (14.3) | 37 (14.9) | |
| Patient safety education | No | 145 (26.4) | 85 (28.3) | 60 (24.1) | −1.12 (.263) |
| | Yes | 404 (73.6) | 215 (71.7) | 189 (75.9) | |

EFA = Exploratory factor analysis; CFA = Confirmatory factor analysis; M = Mean; SD = Standard deviation.

Convergent validity analysis showed that standardized factor loadings ranged from.43 to.73, and the AVE values ranged from.48 to.57, with the AVE for Factor 1 falling below the recommended threshold of.50. The CR values ranged from.80 to.90, with all factors exceeding the acceptable criterion of.70. However, some confidence intervals for the inter-factor correlation coefficients include 1.00, suggesting that discriminant validity was only partially supported (Table 2).

The initial research model (Fig 2A) assumed a structure comprising six first-order latent factors that were allowed to correlate freely. However, discriminant validity was not fully supported due to high inter-factor correlation coefficients.

**Table 2. Result of confirmatory factor analysis (N = 249).**

| Factor | Item | β | B | SE | C.R. | AVE | CR |
|---|---|---|---|---|---|---|---|
| Factor 1 | xa1 | .44 | 1.00 | | | .48 | .82 |
| | xa2 | .59 | 1.64 | 0.27 | 6.05 | | |
| | xa3 | .62 | 1.60 | 0.26 | 6.19 | | |
| | xa4 | .63 | 1.68 | 0.27 | 6.24 | | |
| | xa6 | .58 | 1.55 | 0.26 | 6.01 | | |
| Factor 2 | xb1 | .61 | 1.00 | | | .51 | .84 |
| | xb3 | .63 | 1.00 | 0.12 | 8.36 | | |
| | xb5 | .66 | 1.10 | 0.13 | 8.74 | | |
| | xb6 | .43 | 0.67 | 0.11 | 6.15 | | |
| | xb8 | .60 | 1.01 | 0.13 | 8.12 | | |
| Factor 3 | xc1 | .65 | 1.00 | | | .57 | .90 |
| | xc2 | .71 | 1.03 | 0.11 | 9.84 | | |
| | xc3 | .73 | 1.11 | 0.11 | 10.10 | | |
| | xc5 | .71 | 0.96 | 0.10 | 9.76 | | |
| | xc8 | .67 | 0.96 | 0.10 | 9.29 | | |
| | xc9 | .68 | 0.99 | 0.10 | 9.51 | | |
| | xc11 | .64 | 0.87 | 0.10 | 8.92 | | |
| Factor 4 | xd2 | .67 | 1.00 | | | .52 | .84 |
| | xd4 | .65 | 1.01 | 0.11 | 9.28 | | |
| | xd7 | .59 | 0.90 | 0.11 | 8.49 | | |
| | xd11 | .67 | 1.05 | 0.11 | 9.54 | | |
| | xd14 | .52 | 0.75 | 0.10 | 7.67 | | |
| Factor 5 | xe3 | .65 | 1.10 | | | .51 | .80 |
| | xe6 | .65 | 1.01 | 0.11 | 9.13 | | |
| | xe8 | .63 | 1.10 | 0.12 | 8.90 | | |
| | xe9 | .57 | 0.86 | 0.11 | 8.15 | | |
| Factor 6 | xf2 | .62 | 1.00 | | | .57 | .84 |
| | xf5 | .70 | 1.18 | 0.14 | 8.70 | | |
| | xf8 | .60 | 1.00 | 0.13 | 7.76 | | |
| | xf9 | .62 | 1.01 | 0.13 | 7.96 | | |

| Factor A ⟷ Factor B | Φ | SE | Φ-2.00×SE | Φ+2.00×SE |
|---|---|---|---|---|
| Factor 1 ⟷ Factor 2 | 0.94 | 0.02 | 0.89 | 0.98 |
| Factor 1 ⟷ Factor 3 | 0.91 | 0.03 | 0.86 | 0.96 |
| Factor 1 ⟷ Factor 4 | 0.96 | 0.03 | 0.91 | 1.01 |
| Factor 1 ⟷ Factor 5 | 0.99 | 0.03 | 0.94 | 1.04 |
| Factor 1 ⟷ Factor 6 | 0.82 | 0.02 | 0.78 | 0.86 |
| Factor 2 ⟷ Factor 3 | 0.96 | 0.04 | 0.89 | 1.03 |
| Factor 2 ⟷ Factor 4 | 0.98 | 0.03 | 0.91 | 1.04 |
| Factor 2 ⟷ Factor 5 | 0.96 | 0.03 | 0.89 | 1.02 |
| Factor 2 ⟷ Factor 6 | 0.91 | 0.03 | 0.86 | 0.97 |
| Factor 3 ⟷ Factor 4 | 0.95 | 0.04 | 0.87 | 1.02 |
| Factor 3 ⟷ Factor 5 | 0.95 | 0.04 | 0.88 | 1.02 |
| Factor 3 ⟷ Factor 6 | 0.84 | 0.03 | 0.77 | 0.90 |
| Factor 4 ⟷ Factor 5 | 0.96 | 0.04 | 0.89 | 1.03 |
| Factor 4 ⟷ Factor 6 | 0.90 | 0.03 | 0.84 | 0.96 |
| Factor 5 ⟷ Factor 6 | 0.85 | 0.03 | 0.80 | 0.91 |
| Criteria | Whether [Φ ± 2.00×SE] includes 1.00 | | | |

β = Standardized coefficient; B = Unstandardized coefficient; SE = Standard error; C.R. = Critical ratio; AVE = Average variance extracted; CR = Construct reliability; Φ = Correlation; Factor 1 = Patient identification; Factor 2 = Medication error prevention; Factor 3 = Fall prevention; Factor 4 = Infection prevention; Factor 5 = Life safety compliance; Factor 6 = Information sharing.

When such correlations compromise discriminant validity, it is recommended to either simplify the model to a single-factor structure or specify the factors as mutually independent [25]. Therefore, two alternative models were tested using CFA: Alternative Model I, which specified a unidimensional single-factor structure (Fig 2B); and Alternative Model II, which retained the six-factor structure but constrained all inter-factor correlations to zero (Fig 2C). Unlike the original model that allows correlations among latent factors, Alternative Model II assumes that the six factors are orthogonal, thereby representing them as conceptually distinct and statistically independent constructs.

The model fit indices for Alternative Model I were $\chi^2/df = 1.74$, CFI = .90, TLI = .89, SRMR = .048, and RMSEA = .055 (90% CI [.048,.061]). For Alternative Model II, the indices were $\chi^2/df = 1.68$, CFI = .91, TLI = .90, SRMR = .047, and RMSEA = .052 (90% CI [.046,.059]). Both models met the acceptable criteria for overall fit. A model comparison using the Akaike Information Criterion (AIC) and Bayesian Information Criterion (BIC) revealed lower AIC and BIC values for Alternative Model II, uncorrelated six-factor model provided the best fit to the data (Table 3). Fig 2 shows the standardized factor loadings for the initial and alternative models.

### 3.3. Concurrent validity

Results of the concurrent validity analysis showed that the correlation coefficient with the patient participation scale (21 items) was .91 (p < .001), indicating a statistically significant positive correlation. By subfactor, the correlations were as follows: patient identification .84 (p < .001), prevention of medication errors .80 (p < .001), fall prevention .81 (p < .001), infection prevention .83 (p < .001), compliance with safety in daily life .83 (p < .001), and information sharing .78 (p < .001), all of which showed statistically significant positive correlations.

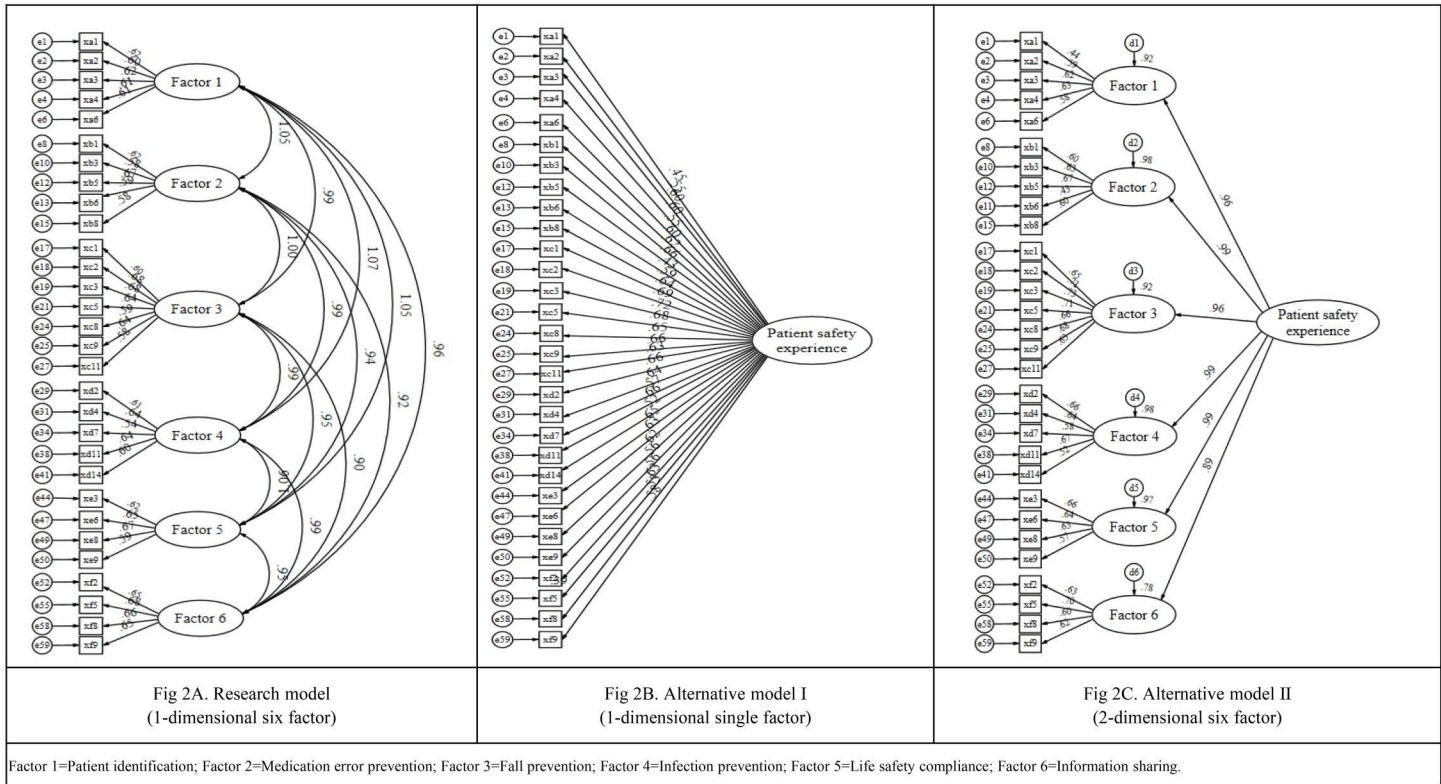

| Fig 2A. Research model (1-dimensional six factor) | Fig 2B. Alternative model I (1-dimensional single factor) | Fig 2C. Alternative model II (2-dimensional six factor) |

Factor 1=Patient identification; Factor 2=Medication error prevention; Factor 3=Fall prevention; Factor 4=Infection prevention; Factor 5=Life safety compliance; Factor 6=Information sharing.

**Fig 2. Research model.**

**Table 3. Model fit results (N = 249).**

| Model | χ² | df | p | χ²/df | CFI | TLI | SRMR | RMSEA (90% CI) | AIC | BIC |
|---|---|---|---|---|---|---|---|---|---|---|
| Research model | 663.48 | 390 | <.001 | 1.70 | .91 | .90 | .047 | .053 (.046~.060) | 813.48 | 1077.29 |
| Alternative model I | 706.79 | 405 | <.001 | 1.74 | .90 | .89 | .048 | .055 (.048~.061) | 826.79 | 1037.84 |
| Alternative model II | 671.71 | 399 | <.001 | 1.68 | .91 | .90 | .047 | .052 (.046~.059) | 803.71 | 1035.86 |

CFI = Comparative fit index; TLI = Tuker-lewis index; SRMR = Squared root mean square residual; RMSEA = Root mean square error of approximation; AIC = Akaike Information Criterion; BIC = Bayesian Information Criterion.

### 3.4. Reliability

The internal consistency of the scale, as measured by Cronbach's α, was.95, indicating excellent overall reliability. The subfactor reliability coefficients were as follows: patient identification (.83), prevention of medication errors (.83), fall prevention (.87), infection prevention (.83), compliance with safety in daily life (.83), and information sharing (.85), all of which exceeded the acceptable threshold of.80.

Test–retest reliability was evaluated using ICCs based on data from 31 inpatients who completed the scale twice at a two-week interval. The overall ICC was.89, confirming high temporal stability. The sub-factors of ICCs were as follows: patient identification (.83), prevention of medication errors (.63), fall prevention (.90), infection prevention (.83), compliance with safety in daily life (.81), and information sharing (.72), demonstrating acceptable to excellent stability across all domains.

### 3.5. Optimization of the scale

The PSES developed in this study was finalized as a 30-item instrument comprising six factors, reflecting a structure of six statistically independent constructs. The scale uses a 4-point Likert response format ranging from 1 ("strongly disagree") to 4 ("strongly agree"). This scale is calculated as an average, with higher scores indicating greater levels of patient-perceived safety experience during hospitalization.

## 4. Discussion

The purpose of this study was to develop a scale to measure the patient safety experience directly experienced and perceived by inpatients. The resulting 30-item instrument, based on six conceptually distinct factors, enables both total and subscale scoring, allowing flexible interpretation in clinical practice. This structural model, commonly used in disciplines such as social sciences, education, and psychology, enables the evaluation of the overall patient safety experience through total scores or domain-specific subfactor scores, providing flexibility in interpretation and application [28]. Additionally, grounded in international patient safety frameworks (OECD, AHRQ, JCI), the scale holds value as both a self-assessment instrument for patients and an evaluation instrument for healthcare providers. It offers a structured approach to monitoring safety practices and has the potential to support the establishment of a patient-centered safety culture.

In the six subfactors of patient safety experience, Factor 1 was "patient identification," which comprised five items and was associated with being informed about identification procedures and cooperating with personal identification processes. These findings aligned with those of Kim and Park [18], and similar domains were found in the nursing safety activity scales for nurses, reinforcing the importance of accurate patient identification [29]. Patient identification is an essential first step in all medical procedures and plays a pivotal role in ensuring safe healthcare delivery. In the study's in-depth interviews, patients acknowledged the potential risks of safety incidents when their personal information was not

properly verified and reported that they actively cooperated with repeated identification checks conducted by healthcare staff. The items "I carefully checked whether the medical staff verified my name and registration number (or date of birth) during medication, examination, or surgery" and "When I received a patient identification bracelet, I checked if the name was correct before wearing it" were retained as final items, allowing for the evaluation of inpatients' experiences with identification practices. Such active patient involvement has been reported as a key factor in improving care quality and reducing patient safety incidents [30]. The final scale incorporated the core standards for patient identification proposed by the Korea Institute for Healthcare Accreditation [7], enhancing its practical value for evaluating identification-related safety behaviors in clinical settings.

Factor 2 was "prevention of medication errors," which comprises five items, such as receiving explanations from medical staffs about the purpose and effects of medications and informing when the injection fluids are not administered. This finding is consistent with that of previous studies involving inpatients, which emphasized patient safety behaviors, such as being informed about drug efficacy, administration methods, side effects, and precautions, as well as notifying healthcare providers in the event of abnormal infusion rates [16,31]. Medication errors represent the second most common type of patient safety incidents in healthcare settings, accounting for 31.9% of reported events, and their incidence continues to increase annually, highlighting the need for effective prevention strategies [32,33]. Accordingly, healthcare professionals must provide patients with comprehensive explanations regarding the purpose, efficacy, and side effects of medications and rigorously implement safety protocols during medication administration [34]. Moreover, a systematic framework should be established to minimize medication errors by incorporating institutional safeguards and patient involvement. Traditionally, prevention efforts have been provider-centered; however, it is becoming increasingly important to actively engage patients in these efforts. Educational programs aimed at improving patients' communication skills should be developed and implemented to empower them to ask questions about their medications and promptly report abnormal reactions. This shift toward patient-centered medication safety could enhance the overall safety culture in clinical practice.

Factor 3 was "fall prevention," consisting of seven items, including receiving fall prevention education and promptly reporting any fall incidents. Previous studies have emphasized the importance of patient education and environmental interventions in preventing falls among inpatients. Safety nursing activity scales also incorporate actions to eliminate external risk factors, such as checking whether bed rails and wheelchair locks are secure [35]. Furthermore, strategies such as fall risk assessment, patient education, requesting staff assistance during movement, and environmental adjustments have been identified as effective fall prevention measures [36,37], which align with the components identified in this study. Falls can lead to severe outcomes, including mortality, and can negatively affect patient outcomes by exacerbating health conditions and prolonging hospital stay [37]. Studies have shown that patients at high risk of falling are influenced by multiple factors, including medication side effects and physical and emotional health conditions, requiring tailored interventions and patient education [36]. However, despite repeated educational efforts, some patients fail to fully recognize the risk of falling. Therefore, hospitals must implement ongoing and structured educational programs to raise patients' awareness of the seriousness of falls and encourage active participation in fall prevention practices.

Factor 4 was "infection prevention," consisting of five items, including checking whether medical staff performed hand hygiene and properly separated and disposed of medical waste. Previous studies have identified the core elements of infection prevention, such as compliance with hand hygiene guidelines, proper medical waste disposal, and respiratory etiquette [8,19], which aligns with the results of this study. In-depth interviews further revealed that patients emphasized the need for strict hand hygiene by medical staff during invasive procedures, as well as detailed guidance regarding visitor restrictions. Such patient education can enhance the awareness of infection prevention and patient safety, serving as an important driver of desirable safety behaviors. Additionally, the prevention of pressure ulcers, which can compromise the patients' quality of life by extending hospital stay and increasing healthcare costs, was considered. Medical institutions recommend integrated strategies for pressure ulcer prevention, including nutritional assessments and scheduled position changes [38]. Accordingly, the item "I changed my body position periodically rather than staying in one position while

awake" was included to reflect this aspect. This suggests that pressure ulcer prevention is not only a key element of infection control but also contributes to patient comfort and overall safety.

Factor 5 was "life safety compliance," consisting of four items, including checking the location of the call bell and notifying staff in the event of medical equipment malfunction. This domain is consistent with that of previous studies that emphasize the importance of refraining from arbitrarily handling medical devices, knowing how to call for help in emergencies, and adhering to fire safety protocols [18,29,35]. Such daily safety compliance plays a crucial role in minimizing environmental risk factors and ensuring the safety of not only patients but also caregivers and visitors. Medical institutions typically provide regular safety education focused on fire prevention, prohibit the operation of medical equipment by nonclinical personnel, and encourage prompt reporting of equipment malfunctions, thereby underscoring the importance of patient safety [39]. In this study, items were developed to reflect safety behaviors that required the active participation of patients, such as identifying the location, using call bells, and recognizing emergency evacuation signage. These findings suggest that life safety compliance not only helps prevent safety incidents but also reinforces the patient's proactive role in maintaining a safe healthcare environment.

Factor 6 was "information sharing," which comprised four items, including asking questions about areas of curiosity and reading educational materials provided by medical staff. In this study, information sharing was conceptualized as extending beyond the simple transmission of information to include patient engagement in treatment decisions and the cultivation of trust-based relationships with healthcare providers. Previous studies have emphasized that effective communication is a fundamental component of patient safety, contributing to the reduction of medical errors and enhancement of care quality [2,17,40]. When a trusting relationship is established, patients are more likely to actively participate in safety-related behaviors [27] and engage in shared decision-making, taking greater ownership of their health management [41]. However, findings from in-depth interviews indicated that patients often struggled to understand the need for their treatment owing to the use of complex medical terminology and brief, task-oriented explanations by healthcare professionals. This communication barrier limits patients' ability to ask questions or seek clarification, an observation supported by earlier studies reporting that time constraints among medical staff can impede patient-centered communication [27,42]. To enhance patient engagement, healthcare professionals should improve communication strategies; and systemic efforts, such as reducing administrative burdens, are required to support effective interactions.

This scale is meaningful because it allows healthcare institutions to identify and reevaluate vulnerable areas in patient safety practices from a patient's perspective. Furthermore, it encourages patient engagement in safety-related behaviors, thereby contributing to the development of patient-centered safety strategies. This instrument may provide foundational data for improving institutional safety protocols and ultimately support the establishment of a culture of safety that centers on patients.

### 4.1. Limitations

This study has several limitations that should be considered when interpreting the findings. First, the sample consisted primarily of young, female, highly educated surgical patients from five hospitals located in Seoul and its metropolitan area, which may limit the generalizability of the findings to broader inpatient populations, particularly older adults or those with chronic illnesses. As these demographic and clinical traits may influence how patients perceive and report their safety experiences—potentially contributing to higher overall scale scores—caution should be exercised when applying the findings to broader populations. Second, although the average score of the developed scale was relatively high (3.32 out of 4), it did not capture safety experiences in special clinical contexts, such as blood transfusions or care for high-risk patients. Third, one item assessing whether medical staff offered patient safety education may be variably interpreted depending on patients' individual characteristics and levels of health literacy. Finally, the confirmatory factor analysis revealed high correlations between latent variables, suggesting that discriminant validity was not fully secured, indicating the need for conceptual clarification of the sub-factors.

## 4.2. Implications and recommendations

Despite the noted limitations, this study has several strengths. This scale was developed based on internationally recognized patient safety frameworks, including those from AHRQ, JCI, OECD, and the Korea Institute for Healthcare Accreditation, thereby ensuring conceptual validity. A rigorous multi-phase process—encompassing expert review, pilot testing, item analysis, exploratory and confirmatory factor analyses—was employed to ensure content and construct validity, as well as internal consistency. The final 30-item instrument consists of six conceptually meaningful subscales, allowing for both total and domain-specific score interpretations, which support its flexible application in various clinical contexts.

The PSES was designed for ease of administration and interpretation, making it suitable for integration into routine inpatient care. It can be administered by patient safety officers, clinical quality teams, or researchers to assess patients' perceived safety experiences during or immediately after hospitalization. Although clinical cutoff scores have not yet been established, the total score (average of 30 items on a 4-point Likert scale) serves as an overall indicator of patient-perceived safety, while subscale scores can help identify specific areas for targeted quality improvement.

To enhance the external validity and broader applicability of the PSES, future studies should include more diverse inpatient populations, such as older adults, individuals with chronic conditions, and those from varied socioeconomic and educational backgrounds. Additionally, intervention-based research is recommended to assess the scale's sensitivity to changes following safety-enhancing initiatives. Development of condition-specific safety education materials and adaptive questionnaire formats (e.g., branching logic based on patient characteristics) may further improve patient understanding and engagement. Lastly, future research should explore the establishment of clinically meaningful thresholds and normative data by patient group, diagnosis, or care unit, and conduct concept analyses to further clarify and refine the theoretical boundaries of patient safety experience.

## 5. Conclusions

This study developed the Patient Safety Experience Scale (PSES), a 30-item, uncorrelated six-factors, to quantitatively assess inpatients' safety-related experiences. Grounded in internationally recognized patient safety frameworks, the scale demonstrated acceptable internal consistency and construct validity through a rigorous development and validation process. Designed for ease of use in clinical settings, the PSES provides a structured and patient-centered approach to capturing safety perceptions during hospitalization. However, as this study focused solely on internal validation, further research is required to establish criterion validity, responsiveness, generalizability, and cross-cultural applicability in diverse healthcare environments.

## Supporting information

**S1 File. Appendix.**
(DOCX)

## Author contributions

**Conceptualization:** On-Jeon Baek, Sun-Hwa Shin.

**Data curation:** On-Jeon Baek.

**Formal analysis:** On-Jeon Baek.

**Funding acquisition:** Sun-Hwa Shin.

**Investigation:** On-Jeon Baek.

**Methodology:** On-Jeon Baek, Sun-Hwa Shin.

**Project administration:** On-Jeon Baek, Sun-Hwa Shin.

**Software:** On-Jeon Baek.

**Supervision:** Sun-Hwa Shin.

**Validation:** On-Jeon Baek, Sun-Hwa Shin.

**Visualization:** On-Jeon Baek.

**Writing – original draft:** On-Jeon Baek, Sun-Hwa Shin.

**Writing – review & editing:** On-Jeon Baek, Sun-Hwa Shin.

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
