## [Decision Letter · Decision Letter 0]

14 Jul 2025

Dear Dr. Shin,

Thank you for submitting your manuscript to PLOS ONE. After careful consideration, we feel that it has merit but does not fully meet PLOS ONE’s publication criteria as it currently stands. Therefore, we invite you to submit a revised version of the manuscript that addresses the points raised during the review process.

We look forward to receiving your revised manuscript.

Kind regards,

Mohd Ismail Ibrahim, MCom.Med

Academic Editor

PLOS ONE

Additional Editor Comments (if provided):

Reviewers' comments:

Reviewer's Responses to Questions

**Comments to the Author**

1. Is the manuscript technically sound, and do the data support the conclusions?

Reviewer #1: Yes

Reviewer #2: Yes

2. Has the statistical analysis been performed appropriately and rigorously?

Reviewer #1: Yes

Reviewer #2: Yes

3. Have the authors made all data underlying the findings in their manuscript fully available?

Reviewer #1: Yes

Reviewer #2: Yes

4. Is the manuscript presented in an intelligible fashion and written in standard English?

Reviewer #1: Yes

Reviewer #2: Yes

Reviewer #1: This paper presents promising and robust findings that could support healthcare providers in evaluating patient safety from the patient’s perspective and integrating this tool into care plans during hospitalization. Moreover, the scale may be valuable for future researchers seeking to assess patient safety holistically in their studies.

Please provide the IRB approval number, as it’s ciphered. please include the ethical approval forms as a supporting document.

It would be valuable to summarize the limitations identified in the study, if available. Also, adding a section of recommendations would help encourage researchers to conduct further interventional studies to enhance the validity of the instrument. Additionally, it may also be beneficial to investigate other aspects related to the patient safety experience that could be included in the scale.

Please see the attached file...

Reviewer #2: This paper presents the development and internal validation of a scale measuring the safety experience perceived by hospitalized patients. The methodology was according to best practices in scale development and validation, the details were profuse and clear for the most part. The developed scale showed excellent reliability, construct validity and temporal stability, and will probably represent an important instrument to improve pacient safety for inpatients. However, some aspects need revision to clarify some points of the manuscript. Here are my suggestions:

1) This paper reports only the internal validation of the scale, so I think it would be appropriate to mention that in the title.

2) The reason to impose an age limit of 65 years was not explained, and this is unexpected because most hopitalized patients are well over 65 year-old. This is a severe limitation of the application of the scale and deverves a justification.

3) Line 123: details should be given on how were the 90 items of the initial scale created.

4) Line 218: I believe the comma between the words "value" and "and" should be deleted.

5) Line 224: the terminology adopted in factor analysis is to name coefficients by loadings, so I think the term factor loadings should be used throughou the text.

6) Line 204-208: It appears to me that this paragraph would be better placed at the beginning of the Results section.

7) Line 215: could the authors explain why the promax rotation was used, instead of an orthogonal rotation. I found this dificult to understand because there are no compelling arguments against the independence of the dimensions.

8) Line 219-223: please review the writing of this paragraph. It presents the criteria for the definition of adequate fit indices, but as it stands might be confused by the results of the analysis of the fit indices.

9) Line 228: please explain the correlation coefficient for discriminant validity.

10) Line 309: I believe that instead of "were 1.00" should be 'include 1.00".

11) Line 318: please explain the two-dimensional 6-factor structure proposed. There was no mention to the two-distinct, separate, undelying latent factors, and figure 2C shows only a single factor. In my opinion, the difference between the research model and model II is that in the former the factors are allowed to be correlated, while in the latter they are assumed to be independent.

12) Still on the same subject, if I am right and model II represents a data structure where the six factors are assumed to be independent, how this aligns with the use of an oblique rotations?

13) Line 388: I believe there is some text missing in the middle of the sentance.

14) Table 1: unless the patient sex was self-reported, instead of Gender the appropriate term would be Sex.

15) Appendix 1. How should a patient score item 16 if the patient never used a wheelchair?

From my point of view, the weak part of the manuscript is the Discussion section, as the results raise a number of questions that were not approached by the authors. Here are my suggestions to improve that section:

16) The sample characteristics should be justified and whether the results may have been impacted by that specific sample should be discussed. This is because the sample characteristics are not what would be expected from an inpatient population. They are mostly young adults, females, most with university educations, surgical patients, the majority with previous hospitalizations. Therefore, the sample characteristics are not those of a population with chronic illnesses that would be expected from a general hospital-based study.

17) A section discussing the limitations and strenghts of the study needs to be included. Some of the the issues were presented in the Conclusions but a better place would be in the limitations.

18) Also, the authors should discuss whether in their opinion this scale is ready to be applied in clinical practice and, in the afirmative, when, to whom and by whom the scale should be applied. In addition, some leads should be given on how to interpret the scale scores, that is, what is a clinical significant score, should subscale scores be interpreted and how.

19) Following the limitations section, suggestions for future research should be proposed. Again, some were presented in the Conclusions section, but a better place would be in the Discussion.

20) Finally, the conclusions should be limited to the interpretantion of the more relevant aspects of the research. I strongly recommend that the authors clearly emphsize that only the results of the internal validation were presented, and further studies evaluating criterion validity, responsiveness, generalizability and transcultural validity are necessary.

**Do you want your identity to be public for this peer review?** For information about this choice, including consent withdrawal, please see our Privacy Policy

Reviewer #1: **Yes: ** Omar Alrfooh

Reviewer #2: **Yes: ** Antonio Gouveia Oliveira

---

## [Author Response · Author response to Decision Letter 1]

13 Aug 2025

<Reviewer 1’s comment>

Point 1: Please provide the IRB approval number, as it’s ciphered. please include the ethical approval forms as a supporting document.

Response: Thank you for your comments regarding ethical considerations. We have provided the IRB approval number and included it in the text.

Line 261-262: Prior to data collection, approval was obtained from the Institutional Review Board (IRB No: SYU 2023-12-008-003).

Point 2: It would be valuable to summarize the limitations identified in the study, if available. Also, adding a section of recommendations would help encourage researchers to conduct further interventional studies to enhance the validity of the instrument. Additionally, it may also be beneficial to investigate other aspects related to the patient safety experience that could be included in the scale.

Response: We appreciate your thoughtful review. In response, we have revised the manuscript by organizing the Limitations and Recommendations sections under clear subheadings.

Line 506-520: Limitations. This study has several limitations that should be considered when interpreting the findings. First, the sample consisted primarily of young, female, highly educated surgical patients from five hospitals located in Seoul and its metropolitan area, which may limit the generalizability of the findings to broader inpatient populations, particularly older adults or those with chronic illnesses. As these demographic and clinical traits may influence how patients perceive and report their safety experiences—potentially contributing to higher overall scale scores—caution should be exercised when applying the findings to broader populations. Second, although the average score of the developed scale was relatively high (3.32 out of 4), it did not capture safety experiences in special clinical contexts, such as blood transfusions or care for high-risk patients. Third, one item assessing whether medical staff offered patient safety education may be variably interpreted depending on patients’ individual characteristics and levels of health literacy. Finally, the confirmatory factor analysis revealed high correlations between latent variables, suggesting that discriminant validity was not fully secured, indicating the need for conceptual clarification of the sub-factors.

Line 539-548: Implications and recommendations. To enhance the external validity and broader applicability of the PSES, future studies should include more diverse inpatient populations, such as older adults, individuals with chronic conditions, and those from varied socioeconomic and educational backgrounds. Additionally, intervention-based research is recommended to assess the scale’s sensitivity to changes following safety-enhancing initiatives. Development of condition-specific safety education materials and adaptive questionnaire formats (e.g., branching logic based on patient characteristics) may further improve patient understanding and engagement. Lastly, future research should explore the establishment of clinically meaningful thresholds and normative data by patient group, diagnosis, or care unit, and conduct concept analyses to further clarify and refine the theoretical boundaries of patient safety experience.

<Reviewer 2’s comment>

Point 1: This paper reports only the internal validation of the scale, so I think it would be appropriate to mention that in the title.

Response: Thank you for your comments on changing the title to fit the overall structure of the paper. Based on the reviewers' comments, the title has been changed as follows.

Revised manuscript (p1): Development and internal validation of the Patient Safety Experience Scale for inpatients

Point 2: The reason to impose an age limit of 65 years was not explained, and this is unexpected because most hopitalized patients are well over 65 year-old. This is a severe limitation of the application of the scale and deverves a justification.

Response: We sincerely appreciate the reviewer's comments and appreciate the in-depth review for the validity of the study. As the reviewer said, patients aged 65 years or older account for a high proportion of hospitalized patients and are a group with high patient safety risks. Nevertheless, the reason for setting the age limit of 65 years in this study is as follows. Cognitive decline frequently occurs in patients aged 65 years or older (Yuan et al., 2021; Chen et al., 2023), and it was judged that this could affect the reliability and validity of self-report measurement tools (Prusaczyk et al., 2017; Nichols et al., 2023). In addition, in the early stage of measurement tool development, it is important to secure internal validity by targeting a group with relatively uniform response ability, so participants of that age group were selected to select a relatively uniform group. Therefore, this study is significant in establishing the basic validity of a patient safety experience measurement tool for adults aged 19-65, and is expected to serve as a foundation for the development of a comprehensive measurement tool including elderly patients in the future. The literature of the study presented in the response is presented below.

Line 192-197: The inclusion criteria were as follows: In order to secure internal validity by targeting a group with relatively uniform response ability, patients aged 19 to 65 years, patients hospitalized for at least 3 days, able to understand the purpose of the study, and who provided informed consent. Exclusion criteria included patients hospitalized on the same day, patients visiting the outpatient clinic, and those with cognitive problems or inability to communicate.

Yuan, L., Zhang, X., Guo, N., Li, Z., Lv, D., Wang, H., ... & Wu, X. (2021). Prevalence of cognitive impairment in Chinese older inpatients and its relationship with 1-year adverse health outcomes: a multi-center cohort study. BMC geriatrics, 21(1), 595.

Chen, P., Cai, H., Bai, W., Su, Z., Tang, Y. L., Ungvari, G. S., ... & Xiang, Y. T. (2023). Global prevalence of mild cognitive impairment among older adults living in nursing homes: a meta-analysis and systematic review of epidemiological surveys. Translational Psychiatry, 13(1), 88.

Prusaczyk, B., Cherney, S. M., Carpenter, C. R., & DuBois, J. M. (2017). Informed consent to research with cognitively impaired adults: transdisciplinary challenges and opportunities. Clinical Gerontologist, 40(1), 63-73.

Nichols, E., Ng, D. K., Hayat, S., Langa, K. M., Lee, J., Steptoe, A., ... & Gross, A. L. (2023). Measurement differences in the assessment of functional limitations for cognitive impairment classification across geographic locations. Alzheimer's & Dementia, 19(5), 2218-2225.

Point 3: Line 123: details should be given on how were the 90 items of the initial scale created.

Response: Thank you for your comments on the initial draft of the questions. We have reviewed and rewritten the drafted parts of the questions.

Line 126-133: Initial items were developed based on the conceptual attributes of the patient safety experience through literature review and in-depth interviews. According to Devellis and Thorpe [19], the more initial items, the better, and they should be more than 50% more than the number of questions in the final scale. Referring to this recommendation, this study composed the number of initial items by component from 12 to 18. A total of 90 items were generated, consisting of 12 items for patient identification, 14 for the prevention of medication errors, 16 for fall prevention, 16 for infection prevention, 14 for compliance with safety in daily life, and 18 for information sharing.

Point 4: Line 218: I believe the comma between the words "value" and "and" should be deleted.

Response: Thank you for your careful review of the sentence structure. We have rewritten it by removing the comma between the words "value" and "and".

Line 227-228: CFA was conducted to evaluate the model fit. The model was considered acceptable when the ratio of the chi-square (χ²) value to degrees of freedom was 3 or less.

Point 5: Line 224: the terminology adopted in factor analysis is to name coefficients by loadings, so I think the term factor loadings should be used throughout the text.

Response: We sincerely appreciate your help in organizing the words that should be written and used in factor analysis. Based on the review comments, we have changed the word that We wanted to name coefficients as loadings to “factor loadings” in general.

Line 232-238: Standardized factor loadings (β) and modification indices (MI) were reviewed to revise the model, if necessary. To verify the convergent validity, the following criteria were applied: standardized factor loading (β) ≥ 0.50, average variance extracted (AVE) ≥ 0.50, and construct reliability (CR) ≥ 0.70 [27]. Discriminant validity was assessed using the confidence interval of the correlation coefficient (Φ ± 2.00 × SE); if the interval did not include 1.00, the constructs were considered distinct, indicating that discriminant validity was established [26].

Point 6: Line 204-208: It appears to me that this paragraph would be better placed at the beginning of the Results section.

Response: Thank you for your feedback on the smooth and concise refinement of sentences and the harmonious connection between paragraphs. Based on your feedback, We have modified it to place the relevant content at the beginning of the research results section so that the connection between sentences flows naturally.

Line 276-278: The survey took approximately 15–20 min to complete. In total, 551 participants responded: 149 via paper-based survey and 402 via online survey. After excluding two incomplete responses, data from 549 participants were included in the final analysis.

Point 7: Line 215: could the authors explain why the promax rotation was used, instead of an orthogonal rotation. I found this difficult to understand because there are no compelling arguments against the independence of the dimensions.

Response: We would like to thank you for your feedback on the Promax rotation method for exploratory factor analysis of data analysis methods. In this study, the Promax method among oblique rotations was selected considering the possibility of correlation between factors during exploratory factor analysis. Since the scale related to patient safety experience includes various sub concepts that are closely connected to each other in actual clinical and organizational contexts, it was judged that there is a high possibility of correlation between them. Orthogonal rotation assumes complete independence (correlation=0) between factors, but various previous studies have reported that factors related to patient experience actually influence each other. In papers on the development of tools dealing with psychosocial and clinical complex concepts, oblique rotation (Promax, Oblimin, etc.) that allows correlation between factors is suggested to be more valid. In fact, scales with multidimensional properties such as patient experience, patient safety, quality of life, and health-related attitudes are often verified on the premise of correlation between factors. Accordingly, it was difficult to accept the assumption that individual factors were 'completely independent', and it was interpreted that various aspects of each factor were realistically connected. Therefore, in this study, we selected rectangular rotation (Promax) that can reflect the relationship between factors based on theoretical, empirical, and statistical grounds, and judged that this is more appropriate for measuring the complex construct called patient safety experience.

Line 221-226: EFA was performed using the maximum likelihood method, as the assumption of normality was satisfied in the item analyses, making it a statistically robust approach. To allow for correlations among latent factors, Promax rotation—an oblique rotation technique widely used in large datasets—was applied, which generates results based on initial orthogonal rotation outcomes [21]. Items with factor loadings below .50 or communalities below .30 were considered for deletion [25].

Point 8: Line 219-223: please review the writing of this paragraph. It presents the criteria for the definition of adequate fit indices, but as it stands might be confused by the results of the analysis of the fit indices.

Response: We sincerely appreciate your feedback on sentences that may confuse readers about the fit indices. We have deleted sentences that may cause confusion by listing multiple indices and criteria in one paragraph, making it difficult for readers to distinguish between results and criteria, or by mixing criteria and actual analysis results, and have revised each index into shorter sentences to clearly distinguish between criteria (cut-off values) and results (observed values).

Line 227-238: CFA was conducted to evaluate the model fit. The model was considered acceptable when the ratio of the chi-square (χ²) value to degrees of freedom was 3 or less. Goodness-of-fit thresholds were defined as follows: SRMR ≤ 0.08, RMSEA ≤ 0.08 (with a 90% confidence interval upper bound ≤ 0.10), CFI ≥ 0.90, and TLI ≥ 0.90. The final model showed acceptable fit indices: SRMR = 0.06, RMSEA = 0.07 (90% CI: 0.05–0.09), CFI = 0.92, and TLI = 0.91 [23]. Standardized factor loadings (β) and modification indices (MI) were reviewed to revise the model, if necessary. To verify the convergent validity, the following criteria were applied: standardized factor loading (β) ≥ 0.50, average variance extracted (AVE) ≥ 0.50, and construct reliability (CR) ≥ 0.70 [27].

Point 9: Line 228: please explain the correlation coefficient for discriminant validity.

Response: Thank you for your feedback that we need to provide an explanation of discriminant validity so that readers can understand it clearly. We have added more information about discriminant validity.

Line 235-238: Discriminant validity was assessed using the confidence interval of the correlation coefficient (Φ ± 2.00 × SE); if the interval did not include 1.00, the constructs were considered distinct, indicating that discriminant validity was established [26].

Point 10: Line 309: I believe that instead of "were 1.00" should be 'include 1.00".

Response: We sincerely thank you for your precise comments and reviews on the discriminant validity. We have revised the relevant sentence. We would like to express my sincere gratitude once again for your review that allows me to see the numerical analysis in more detail.

Line 323-325: However, some confidence intervals for the inter-factor correlation coefficients include 1.00, suggesting that discriminant validity was only partially supported (Table 2).

Point 11: Line 318: please explain the two-dimensional 6-factor structure proposed. There was no mention to the two-distinct, separate, undelying latent factors, and figure 2C shows only a single factor. In my opinion, the difference between the research model and model II is that in the former the factors are allowed to be correlated, while in the latter they are assumed to be independent.

Response: Thank you for your feedback on the model of this study. We acknowledge the confusion caused by the terminology “two-dimensional” in the description of Alternative Model II. Upon your suggestion, we have revised the terminology and clarified the structural assumptions of the model accordingly. To clarify, Alternative Model II does not introduce two distinct, higher-order latent factors. Rather, it maintains the same six first-order latent factors as the initial research model. The key distinction lies in the correlation structure: in the initial model (Fig 2A), correlations among the six factors are freely estimated, whereas in Alternative Model II (Fig 2C), the six factors are specified as mutually independent, with all inter-factor covariances fixed to zero. In this context, the term “two-dimensional” was inappropriately used and has now been corrected throughout the manuscript to avoid misinterpretation. We have replaced it with “uncorrelated six-factor,” and revised the corresponding explanation in both the text and figure legends. We appreciate your attention to this detail, which helped us

---

## [Decision Letter · Decision Letter 1]

27 Aug 2025

Development and interanl validation of the Patient Safety Experience Scale for inpatients

PONE-D-25-31484R1

Dear Dr. Shin,

We’re pleased to inform you that your manuscript has been judged scientifically suitable for publication and will be formally accepted for publication once it meets all outstanding technical requirements.

Kind regards,

Mohd Ismail Ibrahim, MCom.Med

Academic Editor

PLOS ONE

Additional Editor Comments (optional):

Reviewers' comments:

Reviewer's Responses to Questions

**Comments to the Author**

Reviewer #1: All comments have been addressed

Reviewer #2: All comments have been addressed

2. Is the manuscript technically sound, and do the data support the conclusions?

Reviewer #1: Yes

Reviewer #2: Yes

3. Has the statistical analysis been performed appropriately and rigorously?

Reviewer #1: Yes

Reviewer #2: Yes

4. Have the authors made all data underlying the findings in their manuscript fully available?

Reviewer #1: Yes

Reviewer #2: Yes

5. Is the manuscript presented in an intelligible fashion and written in standard English?

Reviewer #1: Yes

Reviewer #2: Yes

Reviewer #1: (No Response)

Reviewer #2: The authors have responded adequately to all my comments and made the appropriate changes in the manuscript that clarified important points in the design and analysis of the research.

**Do you want your identity to be public for this peer review?** For information about this choice, including consent withdrawal, please see our Privacy Policy

Reviewer #1: No

Reviewer #2: **Yes: ** Antonio Gouveia Oliveira

---

## [Editor Report · Acceptance letter]

PONE-D-25-31484R1

PLOS ONE

Dear Dr. Shin,

I'm pleased to inform you that your manuscript has been deemed suitable for publication in PLOS ONE. Congratulations! Your manuscript is now being handed over to our production team.

Kind regards,

on behalf of

Dr. Mohd Ismail Ibrahim

Academic Editor

PLOS ONE